

# Contrasting Mineral Dust Abundances from X-Ray Diffraction and Reflectance Spectroscopy

Mohammad R. Sadrian[1], Wendy M. Calvin[1], John McCormack[1]

[1] Department of Geological Sciences and Engineering, University of Nevada, Reno, Reno, 89557, USA

*Correspondence to*: Mohammad R. Sadrian (msadrian@nevada.unr.edu)

## Abstract

Mineral dust particles dominate aerosol mass in the atmosphere and directly modify Earth's radiative balance through absorption and scattering. This radiative forcing varies strongly with mineral composition, yet there is still limited knowledge on the mineralogy of atmospheric dust. In this study, we performed X-ray diffraction (XRD) and reflectance spectroscopy

measurements on 37 different atmospheric dust samples collected as airfall in an urban setting to determine mineralogy and the relative proportions of minerals in the dust mixture. Most commonly, XRD has been used to characterize dust mineralogy; however, without prior special sample preparation, this technique is less effective for identifying poorly crystalline or amorphous phases. In addition to XRD measurements, we performed visible, near-infrared, and short-wave infrared (VNIR/SWIR) reflectance spectroscopy for these natural dust samples as a complementary technique to determine minerology

and mineral abundances. Reflectance spectra of dust particles are a function of a nonlinear combination of mineral abundances in the mixture. Therefore, we used a Hapke radiative transfer model along with a linear spectral mixing approach to derive relative mineral abundances from reflectance spectroscopy. We compared spectrally derived abundances with those determined semi-quantitatively from XRD. Our results demonstrate that total clay mineral abundances from XRD are correlated with those from reflectance spectroscopy and follow similar trends; however, XRD underpredicts the total amount

of clay for many of the samples. On the other hand, calcite abundances are significantly underpredicted by SWIR compared to XRD. This is caused by the weakening of absorption features associated with the fine particle size of the samples, as well as the presence of dark non-mineral materials (e.g., asphalt) in these samples. Another possible explanation for abundance discrepancies between XRD and SWIR is related to the differing sensitivity of the two techniques (crystal structure vs chemical

bonds). Our results indicate that it is beneficial to use both XRD and reflectance spectroscopy to characterize airfall dust,

because the former technique is good at identifying and quantifying the SWIR-transparent minerals (e.g., quartz, albite, and

microcline), while the latter technique is superior for determining abundances for clays and non-mineral components.

## 1 Introduction

Mineral dust aerosols are lofted from the surface into the atmosphere, mainly in the arid regions of the world, either affecting

the area nearby or traveling long distances for global impacts (Goudie and Middleton, 2006). Suspended mineral particles

affect air temperature by scattering and absorption of incoming sunlight and outgoing long wave radiation (Miller and Tegen,

1998). Mineral dust-radiation interactions (e.g., absorption and scattering) directly modify Earth's radiative balance and energy

budget, consequently contributing to climate change (Tegen and Lacis, 1996; Tegen et al., 1996). Past studies have discussed

that dust particles' distinctive radiative forcing strongly depends on their particle size distribution (PSD) and mineral

composition (Sokolik and Toon, 1999; Sokolik et al., 2001; Ginoux 2017). Atmospheric dust particles contain a diverse mix

of minerals. Such dust is dominantly composed of quartz, carbonates, iron oxides, clays, sulfates, and feldspars (Engelbrecht

et al., 2016; supplement). Therefore, the relative quantity of the various minerals defines the optical properties of these aerosols.

As a common approach, particulate matter deposited by air fall is collected at different geographic locations to determine

mineralogical composition and abundance as well as particle size distribution. Despite the fact that the physico-chemical

properties of minerals have a substantial impact on dust-related radiative forcing, there is no ideal measurement technique for

identifying these properties. To date, X-ray diffraction (XRD) has been frequently used in various research studies as a primary

or complementary technique to measure the mineral content of dust particles (e.g., Caquineau et al., 1997; Kandler et al., 2009;

Engelbrecht et al., 2009b, 2016, 2017; Nowak et al., 2018). For example, Engelbrecht et al., 2017 performed XRD

measurements on 27 dust samples collected from the Arabian Red Sea coast in order to obtain mineralogy and fractional

abundances of minerals. In that study, they found that the dust samples were mainly dominated by quartz, feldspars, micas,

clays, and halite and to a lesser extent by carbonates, iron oxides, and gypsum. While XRD is a powerful technique for

characterizing crystalline phases, it is less effective at measuring poorly crystalline and amorphous phases (Moore and Reynolds, 1997).

In this research, we use visible, near-infrared, and short-wave infrared (VNIR/SWIR) reflectance spectroscopy as a complementary method to obtain mineral identification and abundances. To date, VNIR/SWIR spectroscopy has not been used to study natural dust particle mineralogy; however, it can provide quantitative measurements and identify both amorphous and crystalline phases (Clark, 1999). This approach has been widely used to obtain mineral compositional information in laboratory and remote sensing applications with particular attention to mineral mixtures (e.g., Mustard and Pieters, 1987; Combe et al.,

2008). Reflectance spectra of mixtures are modelled using radiative transfer (RT) theories, such as developed by Hapke (1981), or linear spectral mixing (LSM) (e.g., Ramsey and Christensen, 1998). LSM is employed when a sample reflectance spectrum is simply a linear combination of the constituents' spectra, whereas RT is commonly utilized when materials are intimately mixed, and light is interacting with several minerals resulting in a nonlinear relationship between abundance and spectral feature strength. Since planetary surfaces are mostly composed of intimately mixed minerals with nonlinear spectral

interactions, RT has been found to be an effective way to derive mineral abundances from reflectance spectra measured from spacecraft and in the laboratory (e.g., Mustard and Pieters, 1987, 1989; Hiroi and Pieters, 1994; Lucey, 1998; Cheek and Pieters, 2014; Robertson et al., 2016; Lapotre et al., 2017). Additionally, many studies have employed RT to model reflectance spectra of synthetic or laboratory mineral mixtures, validating the derived abundances. For example, Robertson and Milliken (2016) demonstrated that physical mixtures of clay and sulfate at varying abundances were accurately determined (within 5

%) using a Hapke RT model.

Here, we used both XRD and reflectance spectroscopy as complementary techniques to investigate the variation of both mineral composition and abundance in natural samples of atmospheric dust deposited in Ilam city, Iran. We estimated mineral abundances of these homogenous samples using their reflectance spectra, a Hapke RT model (RTM) combined with linear

mixing, and compared those results with semi-quantitative abundances determined by XRD. We examined the ability of widely



used spectral mixing approaches to determine if they can be used accurately to quantify mineral abundances in dust samples collected in urban settings.

## 2 Methods and Material

### 2.1 Sample Collection

For this study, we conducted measurements on 37 samples of dust captured with marble dust collectors (MDCO), located in Ilam city, Iran. Based on an original design by Ganor (1975), we chose MDCO due to the efficiency and popularity in desert research (e.g., Offer et al., 1992; Goossens and Offer, 1994; Goossens and Rajot, 2008). In general, the representation of dust in the sample depends on the selected sampling method, which may result in underestimation or overprediction of some important minerals (von Holdt et al., 2021). MDCO (like many other dust catchers) is less efficient in dust collection in high

wind regimes (Goossens, 2005). However, it was proven to be efficient at collecting dry deposition and less sensitive to local weather conditions (Goossens and Offer, 1994; Sow et al., 2006; Goss et al., 2013). Sadrian et al. (2012) selected Ilam city as their study area, because it is located in western Iran and is affected by large dust sources in neighbouring countries including Iraq, Kuwait, and Saudi Arabia (Shahsavani et al., 2012), and thus it is commonly impacted by severe dust storms. To collect deposition of airborne dust, 13 dust samplers were distributed and installed throughout the city area (Fig.1). Deposited dust

was collected in three intervals from September, 2011 through June, 2012 (Appendix A, Table A 1). Specific three-month periods were September 23 to December 21, 2011 (Fall) and December 22, 2011 to March 19, 2012 (Winter), and March 20 to June 20, 2012 (Spring). A total of 39 samples were collected in order to determine their mineralogy, heavy metal content, and deposition rate in different areas of Ilam city (Sadrian et al., 2012). In the current research, we revisit the compositional information of all dust samples that were collected from Ilam city in the time frame of between September 2011 to June 2012.

Since two samples did not contain enough dust for our analysis, as shown in Table A 1, the measurements for this study were conducted on 37 samples.





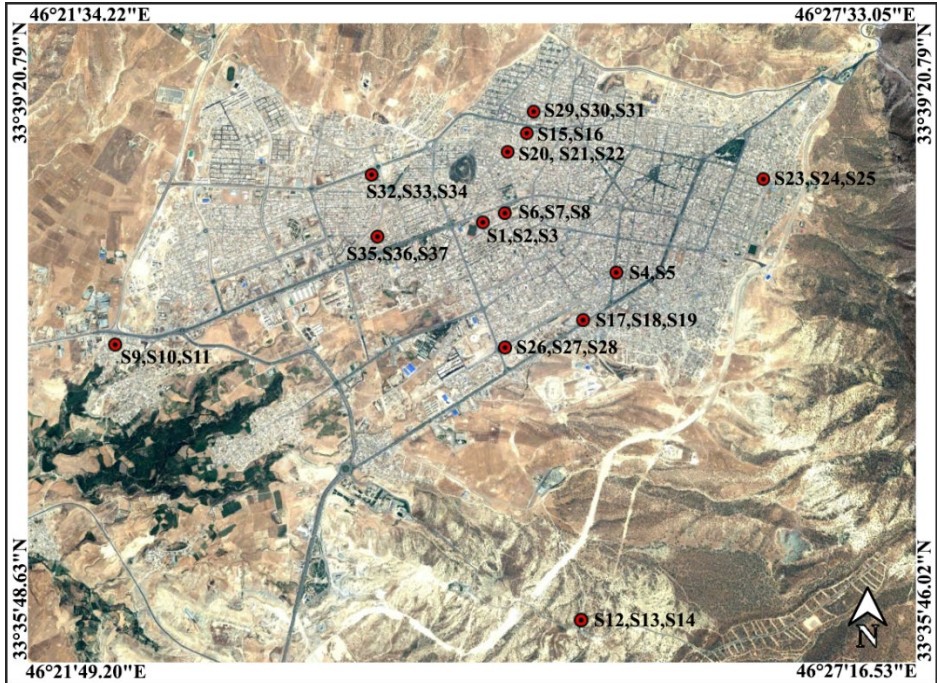

**Figure 1. Map (©Google Earth) shows the distribution of samplers throughout Ilam city. Annotations note sample numbers identified**
**in Appendix A, Table A 1.**

### 2.2 X-Ray Diffraction (XRD)

XRD is a technique used to obtain the unique crystal structure of a material. Diffracted beams are measured over a range of angles (2-theta) and peaks at specific angles are related to the crystal structure of the mineral (Klein, 2002). For the Ilam samples we used a Bruker D2 Phaser benchtop X-ray diffractometer. Qualitative phase identification was performed using

XRD evaluation software (DIFFRAC.EVA), that helps to identify phases in a specimen by comparison with standard patterns existing in a library. Figure 2 displays standard reference minerals with unique diffraction patterns extracted from an accessible, established dataset (American Mineralogist Crystal Structure Database (AMCSD) (Downs and Hall-Wallace, 2003)) compared with unknown peaks in an Ilam sample (S11). As shown in Fig. 2, matches for quartz (Q), calcite (C), albite (Al), microcline (M), gypsum (G), kaolinite (K), and actinolite (Ac) (representative amphibole) were found in S11. Although there were no

standard reference patterns in the AMCSD dataset to show the match peak for illite, we detected this mineral in S11 based on the visual assessment and past published data on the location of illite peaks (Gualtieri, 2000; Drits et al., 2010). Montmorillonite



was easily identified in all samples using spectroscopy (Fig. 3). However, in XRD plots it is difficult to discriminate without special sample preparation. In order to account for montmorillonite, we included the standard reference pattern in all diffractograms and mineral abundance determinations. Semi-quantitative (S-Q) assessment of mineral abundances was

obtained through integrated band area ratios and relative intensities of several lines after removing background and source peak noise. The result from S-Q analysis of all dust samples is discussed in Sect. 3.1.

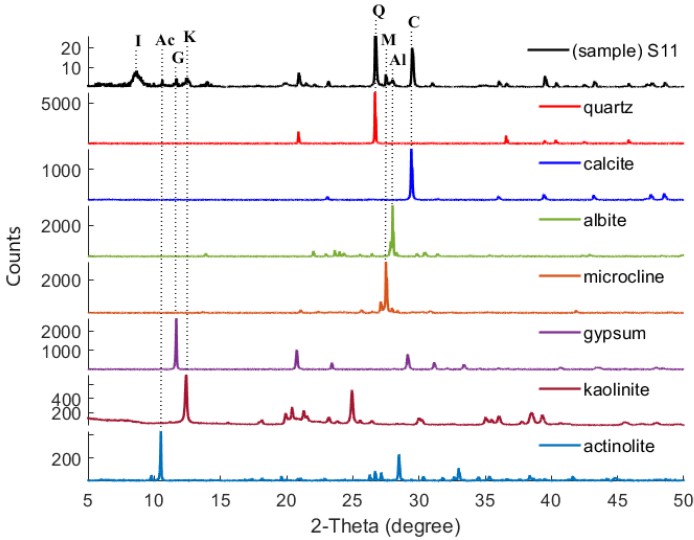

**Figure 2. XRD pattern of sample S11 is compared with those of standard reference minerals from AMCSD. Dotted lines connect the diagnostic XRD peak in quartz (Q), calcite (C), albite (Al), microcline (M), gypsum (G), kaolinite (K), and actinolite (Ac) to the**
**corresponding XRD patterns in S11 confirming the presence of these minerals in this particular dust sample.**

**2.3 VNIR/SWIR Reflectance Spectroscopy**

Minerals have distinctive spectral characteristics, and band center, strength, shape, and width are utilized to confidently identify species (Gaffey et al. 1993; Clark, 1999). In VNIR/SWIR (350 to 2500 nm) diagnostic absorption bands arise from transition electrons (generally caused by iron oxides) in various crystallographic sites and from the overtones and combinations of the

fundamental vibrations of species such as hydroxyl, water, and carbonate (Hunt, 1977; Clark et al., 1990). VNIR/SWIR reflectance measurements of dust samples were carried out by means of a high resolution/high sensitivity field/lab Spectral Evolution (SE), model RS-5400 portable spectroradiometer. To collect sample spectra, dust samples were placed in a holder and a contact probe with a halogen light source was used to capture VNIR/SWIR data. As part of routine calibration, the

contact probe measures a white halon plate. All sample measurements are automatically ratioed to the halon calibration target

resulting in a measurement that is in reflectance. Sample spectra were measured with a 0° incidence angle and a 38° emergence

angle, yielding a 38° phase angle. Because sample volumes were small and to minimize the effect of the aluminium holder

reflectance, we measured the sample on a holder covered with black tape. Measurements of the tape alone confirmed there

were no features introduced by this method.

Mineralogy for the reflectance spectra of the Ilam dust samples was determined by comparing the samples with the well

characterized USGS library (Kokaly et al., 2017). Mineral constituents were identified with an iterative procedure and

inspection where phases were selected on the basis of $H_2O$, OH, and Al-OH absorption features for phyllosilicates, the $H_2O$

band in sulfates, and $CO_3$ in carbonates (Hunt, 1977; Gaffey, 1986; Clark et al., 1990). Figure 3 shows representative spectra

from three Ilam samples (S15, S25, S26) having varying mineralogy. These samples (S15, S25, S26) represent a range of

mineral compositions including calcite, montmorillonite, illite, and gypsum. In this spectral range (1350-2500 nm), common

silicates (dotted flat lines in Fig. 3) microcline, quartz, and albite have no absorption features and are thus known as transparent

minerals (Clark, 1999). We truncated all spectral plots at 1350 nm in order to focus on the above 1350 nm spectral range with

the strongest features. These samples do not include iron oxides, therefore exclusion of the spectral range from 350 to 1350

nm will not miss any major mineral components.

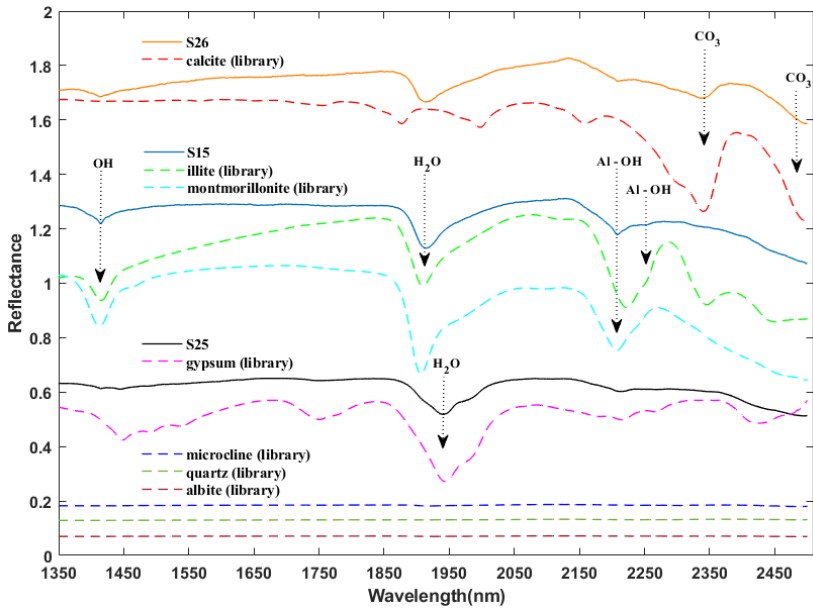


**Figure 3. SWIR spectra for three representative samples (S15, S25, S26) and library spectra of pure minerals showing diagnostic features for calcite, montmorillonite, illite, and gypsum. All spectra are offset for clarity. Arrows near 1400, 1900, 2200, and 2250 nm call out features arising from OH, water, and Al-OH in mineral structures, common to many clay minerals such as montmorillonite and illite. Arrows targeting 2340 and 2480 nm show the wavelengths of dominant absorption features in calcite.**
**Arrow at 1945 nm represents the unique spectral signature attributed to water in sulfates such gypsum.**

## 2.4 Optical Microscopy (OM)

An Olympus petrographic optical microscope was used to assess mineralogical composition and relative abundance of minerals in the samples. Mineral grains were mounted on a glass slide immersed with Cargille 1.544 refractive index oil. Particles were identified based on their diagnostic properties such as color, cleavage, refractive index, and texture. We were able to detect
some coarser particles such quartz, carbonates, and amphibole (Fig. 4a and 4b), however, fine grain clay minerals were not identifiable due to the petrographic microscopy limitation for grain sizes less than 10 µm. The presence of manmade materials (which could be related to asphalt and tar) was revealed by visual inspection of OM images. Figures 4a and 4b illustrate the relative abundances of these dark materials in samples compared to mineral particles. Additionally, there are numerous angular particles, particularly in Fig. 4b, that have no cleavage, a refractive index significantly lower than 1.544, no crystal structure,



and seem to be amorphous and isotropic. Because these samples were collected in an urban setting, they contain a variety of different anthropogenic particles that are difficult to identify using OM.

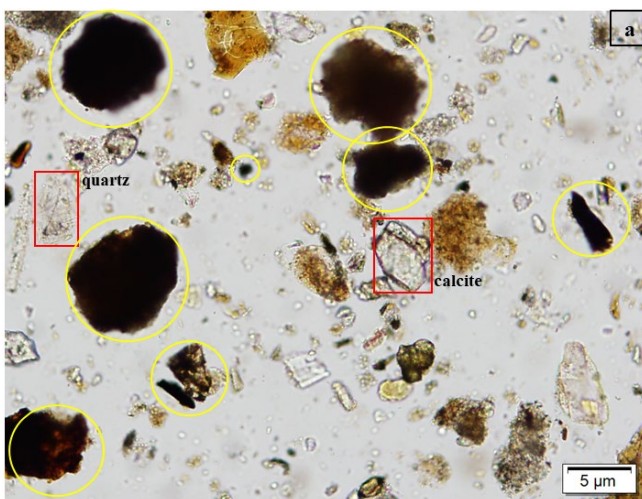
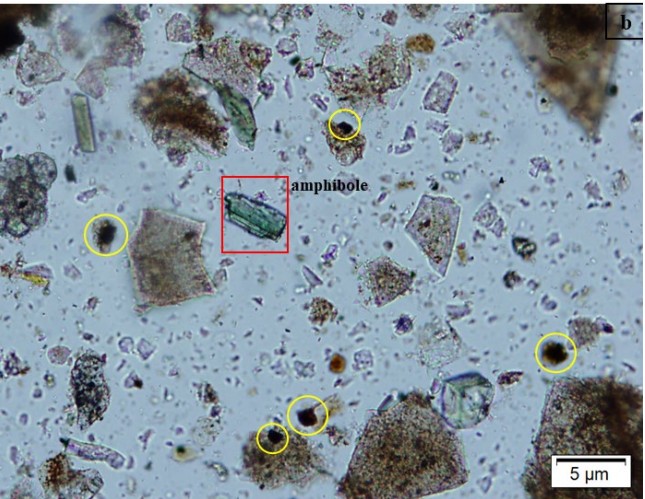

**Figure 4. (a) and (b) are OM images of representative samples S6 and S11, respectively, depicting the presence of quartz, calcite, amphiboles (red rectangles), and dark materials (yellow circles). Quantitative and visual assessment reveals that both images contain a higher abundance of dark materials and other unknown particles, that have no diagnostic mineral properties.**

## 2.5 Particle Size Distribution (PSD)

PSD was determined for all Ilam samples using a Malvern Mastersizer 3000. This instrument is based on a compact optical

system that uses laser diffraction to measure particle size distribution for both wet and dry dispersions (known as hydro and aero methods). We selected the hydro method for PSD analysis because this technique further disperses the sample by sonication for 120 seconds. Wet measurement offers the advantage of being able to measure for a longer periods of time, allowing for a complete measurement of the entire size distribution. This method also allows for full sample recovery. For subsequent analysis, the particle size fractions that make up the samples were categorized into three groups: clay ($< 2$ μm), silt

(2-63 μm), and sand (63-500 μm). The 37 dust samples were dominated by silt sizes but showed variable size range distribution, as shown in Figure 5. The modes for each size ranges are clay ~ 6 %, silt ~ 85 %, and sand ~ 3 %.





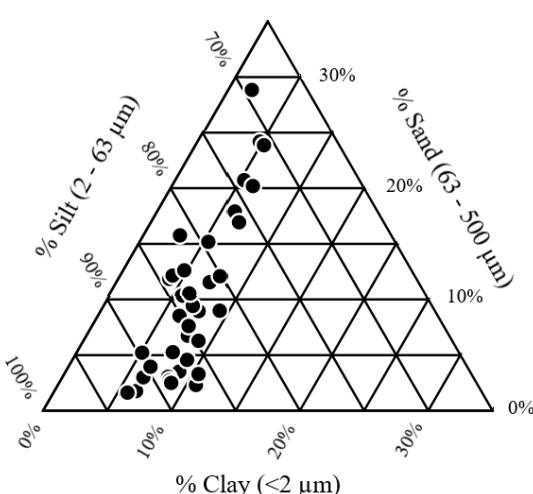

**Figure 5. Ternary diagram showing volume distributions for 37 dust samples analyzed with Malvern Mastersizer 3000. Silt is the most prevalent size class in the samples with min ~ 69 % and max ~ 93 %.**

**2.6 Mineral Abundance Estimation from Reflectance Spectra**

In order to determine mineral abundances from reflectance spectra, we initially used linear spectral mixing (LSM) of the reflectance spectra. This approach assumes that the spectrum of the sample is a linear combination of the spectra of individual minerals (endmembers) and it has been extensively used to characterize materials on the surface of Earth (e.g., Metternicht and Fermont, 1998; Roberts et al., 1998; Dennison and Roberts, 2003) and Mars (e.g., Bell et al., 2002; Combe et al., 2008).

Based on LSM, the reflectance spectra of a mixture can be expressed as (Keshava and Mustard, 2002),

$$Y(\lambda) = \sum_{i=1}^{n} \alpha_i X(\lambda)_i + \varepsilon(\lambda) ,$$

(1)

where Y denotes the reflectance for the mixed spectrum, $\alpha_i$ is the abundance of the $i$th endmember in the mixture spectra, $\lambda$ is the wavelength, $\varepsilon$ represents the residual error between sample and modelled spectra, and $X(\lambda)$ is the matrix of input endmembers reflectance spectra obtained from the USGS spectral library (Kokaly et al., 2017).


To solve equation 1 for $\alpha_i$, we employed a non-negative linear least squares (NNLS) algorithm which calculates a component's coefficient or abundance, which must be a positive number (Rogers and Aharonson, 2008). Our NNLS algorithm was designed





using Matlab R2019a and an available function called non-negative linear least squares (lsqnonneg). The inputs for the NNLS model are the mixture reflectance and the matrix of endmember reflectance spectra from the USGS library, and outputs are

the vectors of abundances and the root mean square error (RMSE) between the sample spectra and the model fit. In order to assess the quality and the accuracy of the modelled spectra, both visual comparison of the calculated fit and the root mean square error (RMSE) were evaluated. While application of this method resulted in a very low RMSE for the fit between the sample and modelled spectra, the modelled spectra did not match band centers and strengths for the absorption features and did not produce reasonable mineral abundances. As these samples are very fine-grained, with an intimate association with one

another, multiple scattering effects are expected to be important, and thus reflectance spectra of the mixture are a nonlinear combination of constituents' abundances (Nash and Conel, 1974; Singer, 1981). In order to address this nonlinear mixing, we implemented a widely used radiative transfer model based on Hapke (1981) that has been shown to provide reliable mineral abundances from laboratory particulate mixtures (e.g., Mustard and Pieters, 1987, 1989; Hiroi and Pieters, 1994; Lucey, 1998; Robertson et al., 2016; Lapotre et al., 2017). In order to determine abundance, the mixture and library mineral endmember

reflectance spectra are converted to single scattering albedo (SSA) according to Equation 2 (Hapke, 1981, Eq. (37)). SSA is the ratio of the scattering to the extinction of the medium. A combination of the SSA of mineral endmembers do mix linearly [Johnson et al., 1983] and thus is able to accurately reproduce the mixture reflectance spectra. Mixture reflectance spectra are related to the average SSA ($w$) through

$$r = \frac{w}{4} \; \frac{1}{\mu_\circ + \mu} \{[1 + B(g)]P(g) + H(\mu_\circ)H(\mu) - 1\}, \tag{2}$$

where $r$ is the reflectance, $\mu_\circ$ and $\mu$ are the cosines of the angles of incident and reflected light, $w$ is the average single scattering albedo, $H$ is the Chandrasekhar function for isotropic scatterers, $B(g)$ is backscatter function, $P(g)$ is the average single particle phase function, and $(g)$ is phase angle. Following the reasoning of Mustard and Pieters (1989) that there is negligible backscattering at intermediate phase angles, we set backscatter function B(g) to zero. We assume these particles scatter isotropically and we can set $P(g) = 1$. Hapke's approximation of Chandrasekhar's $H$ function is defined by Eq. 3,

$$H(\mu) = \frac{1 + 2\mu}{1 + 2\mu\gamma} \; , \tag{3}$$

Where $\gamma = \sqrt{1 - w}$. We now invert Eq. 1 to calculate $w$ based on the reflectance measurement, which yields the expression,





$$w = \frac{4(\mu+\mu_{\circ})r}{H(\mu)H(\mu_{\circ})} \; , \tag{4}$$

where we use Eq. 3 to obtain $H(\mu)$ and $H(\mu_{\circ})$. This equation includes $w$ on both the left side and in the $H$ functions. In order to solve this, $w$ is subtracted from both sides of the equation and we solve for the value of $w$ that results in zero, using the

Matlab command "fzero". This command is used to find the roots for nonlinear equations of a single variable. As $r$, $\mu$ (38°), and $\mu_{\circ}$ (0°) are known, the root is the value of $w$ that makes the whole equation zero. Using this method, we derived single scattering albedo at each wavelength for both dust mixture spectra and the mineral endmembers from the library. Because the average SSA ($w$) of a sample is a linear combination of individual mineral SSA, we employ a linear spectral mixing approach (Eq. 1), but Y is now $w$ of the measured sample and X is the SSA spectra of pure minerals from the USGS library. Using Eq.

5, we determine the fractional contribution of a given mineral.

$$w_{mix} = \sum_{i=1}^{N} f_i \, w_i \; , \tag{5}$$

where $w_{mix}$ is the average SSA, $w_i$ is the SSA for individual endmember $i$, and $f_i$ is the fractional geometric cross-section for component i. Based on Lapotre (2017), $f_i$ can be expressed as,

$$f_i = (\frac{m_i}{\rho_i d_i})/\Sigma(\frac{m_n}{\rho_n d_n}) \tag{6}$$

for an n component mixture. In Eq. 6, $d_i$, $\rho_i$, and $m_i$ are the grain size, density, and mass abundance of endmember $i$. Past studies reported the density of dust particles between 2 to 3 g cm$^{-3}$ (e.g., Delany et al., 1967; Maring et al., 2000; Reid et al., 2003; Fratini et al., 2007), so we set the density as 2.5 g cm$^{-3}$ for all dust samples. Based on information provided in the USGS library, we selected spectra measured at finer grain sizes when available. The USGS library includes multiple samples for a

given mineral type. Through trial and error, we selected individual samples that provided the best fits. These spectra are shown in Appendix B (Fig. B1). Most library minerals used were in the grain size (or diameter) range <150 μm, but our model assumes all components have the same grain size and does not allow this to vary as a free parameter.

To derive fractional abundances, the NNLS Matlab solver is used to input a matrix of mineral endmember SSA and dust sample

$w$. This algorithm attempts to find the mass abundances that reproduce the best model fit for a sample spectrum. Figure 6 displays the calculated linear least squares fit of the model to the measured spectra of three representative dust samples (S15,



S33, S1). In addition to minerals, we found hydrocarbon (C-H) absorption features related to asphalt and tar in many of the samples in our preliminary analysis, and thus we included their spectra (Fig. 7a) in the input endmember bundles for modelling of all 37 samples. Our analysis determined the RMSE between the sample and the modelled spectrum with variable small

numbers between 0.058 to 0.28. Sample S15 (Fig. 6a) displays a well-modelled fit based on our visual evaluation and a low RMSE (0.058). Many of the samples, such as S33 (Fig. 6b), used a substantial amount of asphalt or tar to reproduce a good fit in the wavelength region between 2300-2360 nm.  Some parts of the fit for S15 and S33 have discrepancies (e.g., at 2255 nm for S15, and between 1650-1850 nm for S33), but absorption feature shapes and centers are accurately determined. We found many spectra (S1, Fig. 6c) are not well modelled due to the contribution and presence of other materials. We visually identified

dry grass, plastic, and styrofoam in some samples. Figure 7b shows the spectra for dry grass and plastic extracted from the USGS library (Kokaly et al., 2017), as well as styrofoam that we characterized in the laboratory. These urban materials have strong absorptions with a wide range of spectral features. These materials are often not part of existing spectral libraries and not including them in the model likely prevents a good match to the measured spectra. Because the focus of this research was on the mineral constituents, we did not attempt to include other non-mineral components in order to obtain good fits for all

samples. Asphalt and tar were included in all models because their absorption bands occur in many samples and provide good match to the overall SSA.

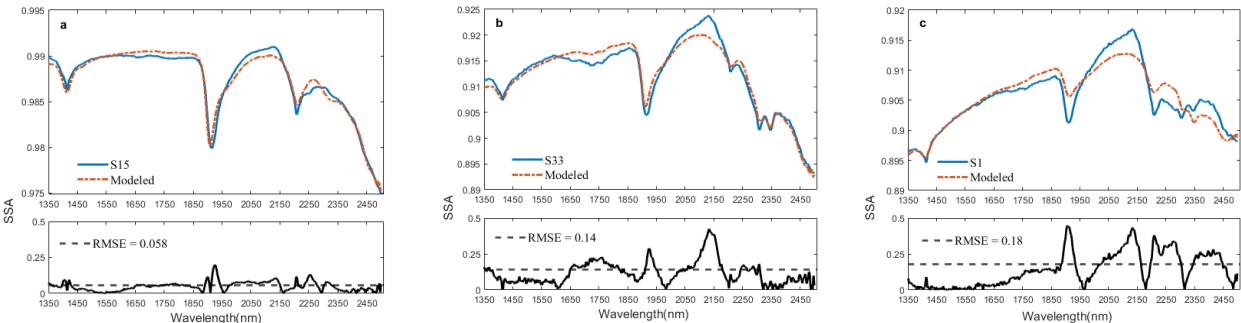

**Figure 6. 6a, 6b, and 6c display the model fit for the representative samples S15, S33, and S1. Measured spectra are shown with solid**
**blue lines and modelled with dash-dot red lines. The smaller plots on the bottom show the root mean square error (RMSE) as a function of wavelength and the total RMSE. The fit uses SSA derived from library endmembers reflectance spectra. Out of 37 modelled spectra, S15 (a) shows the best fit and the lowest RMSE; however, S1 (c) shows misfits and one of the highest RMSE. Materials contributing to the misfits are discussed in the text.**





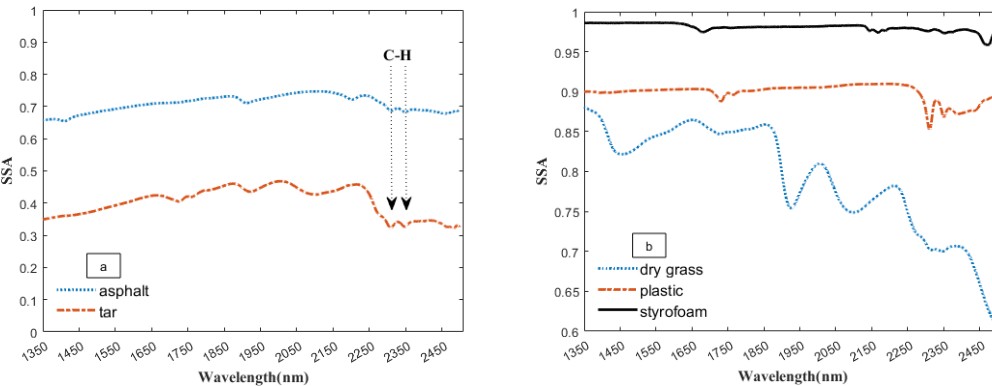

**Figure 7. Plots (a) and (b) show the spectra for non-mineral materials common in urban settings. The arrows in plot (a) point to a doublet arising from C-H bonds in asphalt and tar. Spectra in (b) are for other materials that were visually identified in the samples whose absorption features may lead to poorer model fits. All spectra are offset for clarity.**

## 3 Results

### 3.1 XRD (Total Mineral Abundance)

S-Q analysis, as described in Sect. 2.2, resulted in mineral mass abundances shown in Fig. 8. The XRD bar chart (Fig. 8) indicates that individual mineral abundances vary from sample to sample, yet there is some regularity. Quartz and albite (plagioclase), followed by illite (clay) are the most common minerals in the samples. Kaolinite and montmorillonite (clays) are dominantly detected in minor and trace levels in the samples, and thus make up a small fraction of the total abundances. Some minerals in the XRD bar chart are more variable both in their presence and abundance. Calcite (carbonate) show the

highest variation with a range between 0-63 % of the total mineral abundance. Microcline (K-feldspar), actinolite (amphibole), and gypsum (sulfate) are among the least common minerals. XRD detected gypsum in only three samples collected close to construction sites. Since this mineral is a common mineral on many construction sites, its infrequent and rare presence may be derived from nearby building materials.





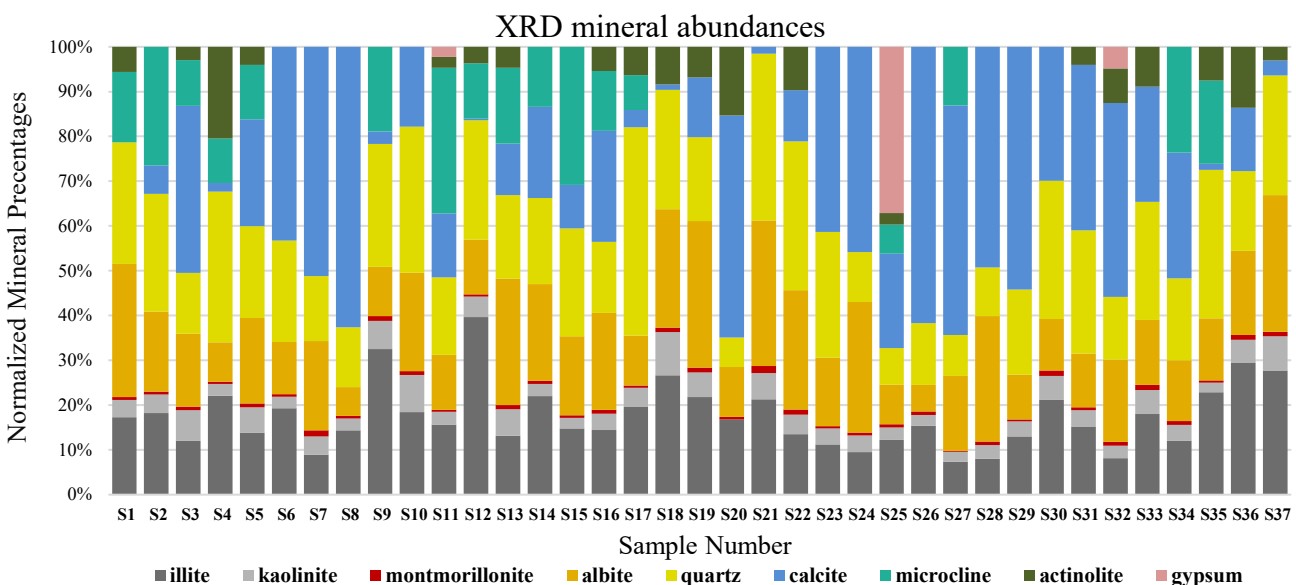

**Figure 8. Bar charts demonstrate the relative phase concentration (%) calculated from the total diffracted peak area of various minerals obtained by XRD analysis.**

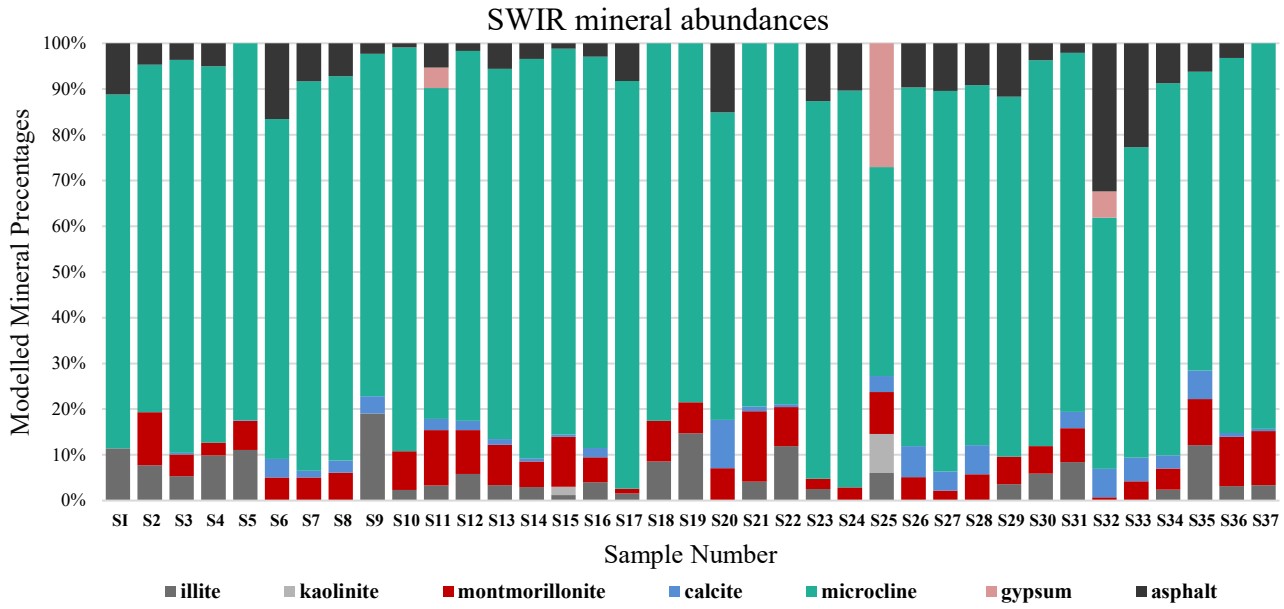

**Figure 9. Bar charts shows the relative mass concentration (%) calculated from a linear combination of SSA of minerals and asphalt. The unrealistically large proportion of microcline is discussed in the text.**





## 3.2 VNIR/SWIR Reflectance Spectroscopy

As discussed in Sect. 2.6, all spectra were modelled to derive mineral abundances. Goodness of the fit is highly dependent on the input endmembers. While additional endmembers can improve the quality of the model, incorporating extra endmembers just to improve the fit can lead to erroneous abundances. Therefore, we included only the phases that were identified with SWIR and XRD based on diagnostic features. Figure 9 demonstrates mineral abundance variations obtained from linear mixing of SSA. This figure depicts high abundance of microcline, although this mineral (along with quartz and albite) is featureless in the SWIR range (Fig. 3). As also shown in Fig. 3, pure library minerals have much stronger absorption features (greater depth) than observed in the Ilam samples. This is referred to as higher spectral contrast. Therefore, the model automatically uses microcline as a neutral endmember to create a model spectrum that fits weaker features. By incorporating featureless material, the overall spectral contrast is reduced at all wavelengths (Hamilton et al., 1997 and 2000). This results in relatively low abundances of other minerals such as illite and calcite (Fig. 9). In order to better compare to XRD, we removed microcline and other spectrally neutral minerals and then re-normalized the abundances for both XRD and SWIR (Fig. 10).

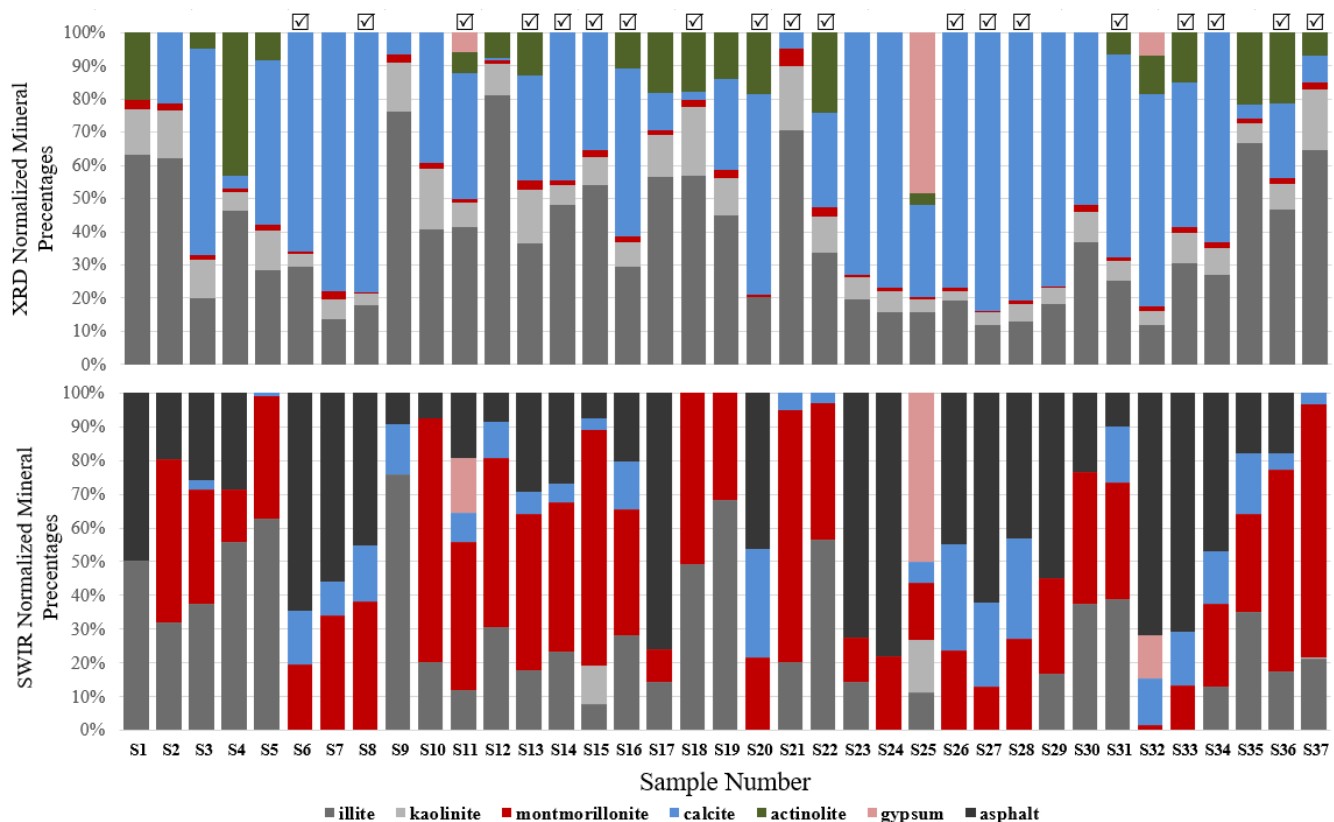

**Figure 10. Bar charts show XRD (top) and SWIR (bottom) normalized abundances after removing the transparent minerals. Those samples with check marks had good spectral model fits as described in the text.**



Figure 10 displays normalized mineral fractions (%) after removing microcline from the SWIR and removing quartz, albite, and microcline from the XRD. Comparison of these bar charts reveals that SWIR models are dominated by the abundance of clays and asphalt, with a low abundance of carbonate. Montmorillonite is the most prevalent clay mineral found in the dust mixtures with highly variable illite, and kaolinite is only seen in a few samples. Since it is difficult to distinguish montmorillonite from illite using XRD, we will compare abundance of all clay minerals in the next section. Surprisingly,

asphalt has a high fraction and is included in the models of the majority of samples, but would not be observed by XRD due to its lack of crystal structure. This suggests that asphalt may act as an agent to reduce spectral contrast and contributes to the lower relative abundance of carbonate, similarly to that of microcline. The three samples that contain gypsum are the same in both SWIR and XRD. Although actinolite (amphibole) is a variable component in the XRD data, it is not apparent or used in the SWIR models.

**3.2 Comparison of Mineral Abundances from XRD and SWIR Spectroscopy**

Due to the contribution of the non-mineral materials in the samples, many model fits were poor (e.g., Fig. 6c) and hence did not retrieve mineral abundances correctly. Poor models may omit, underestimate, or even overestimate the abundance value for specific minerals. In order to better compare the mineral abundances derived from the spectra and XRD S-Q results, a thorough examination of model fit was carried out. Inspection of both model fit match quality and RMSE identified 19 samples

that had well-matched absorption feature centers and strengths (check marks in Fig. 10). In order to compare equivalent abundances, transparent mineral amounts were first removed from both SWIR and XRD (Fig. 10), and then endmember fractions that had non-zero values were re-normalized to 100 %. Illite and kaolinite are among the most common minerals detected with XRD, but in SWIR, the former displays a very high variability, and the latter only exists in two samples. Montomorillonite presents as a small fraction in XRD abundances but is often quite high in SWIR. In order to compare illite,

kaolinite, and montmorillonite abundances from XRD and SWIR, we collected their abundances together into a clay group. Figure 11 compares the abundance of the dominant mineral components (clays and carbonates) for the 19 samples having good spectral fits. Figure 11 demonstrates a positive correlation with a high correlation coefficient for both clay and carbonate abundance values from XRD and SWIR. However, where the best fit correlation for clays displays a linear relationship between abundances generated from these two approaches, the one-to-one comparison of the fractions generally shows an

underestimation of the amount of clay by XRD. On the other hand, the best fit correlation plot for calcite (Fig. 11b), despite having a high r-squared ($R^2 = 0.75$) and visually good correlation, indicates that SWIR significantly underestimates calcite abundances compared to the corresponding XRD percentages.





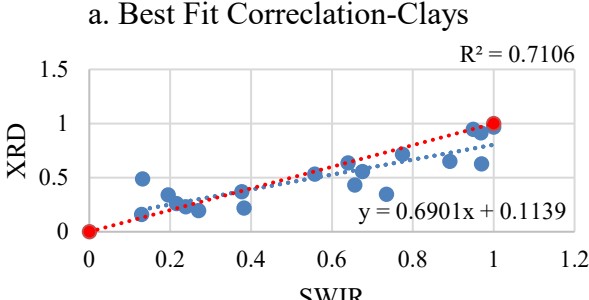

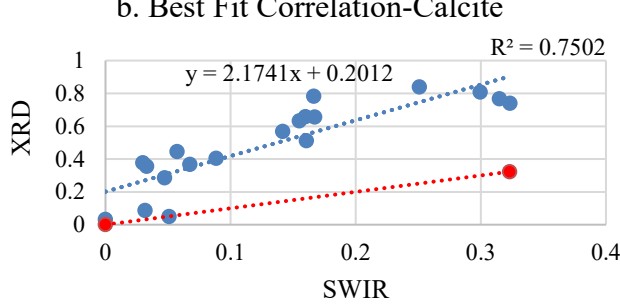

**Figure 11. Plots display the difference between abundance values derived from XRD and SWIR, data and the best fit line are in blue, the 1:1 correspondence is in red. In (a) clay abundances obtained from SWIR demonstrate a relatively higher value suggesting underestimation by the XRD. In (b) the SWIR derived abundances strongly underestimate the amount of calcite compared to XRD.**

## 4 Discussion - Discrepancies in Derived Abundances between XRD and SWIR

In this study, we obtained compositional information and mineral mass abundances for dust samples from both XRD and SWIR. The goal was to compare spectrally derived abundances with S-Q determined abundance values via XRD. We also aimed to evaluate if combining the Hapke model for SSA and the LSM can accurately predict mineral abundances in natural dust samples collected in urban areas. SWIR vastly overpredicts microcline abundances as this spectrally neutral mineral is automatically employed in modelling to uniformly decrease spectral contrast between measured spectra and model fit. After normalizing both data sets for the influence of transparent minerals on the SWIR data, our analysis illustrates that XRD somewhat underpredicts total clay mineral content (Fig. 11), but underpredicts montmorillonite by a significant margin (Fig. 10). In contrast, spectral modelling predicts a considerable amount of montmorillonite (up to 75 %) in the samples. Comparison of clay abundances from these two techniques showed a positive correlation. However, individual sample comparisons generally showed a higher abundance for clays derived from spectroscopy. Calcite abundances determined from XRD and SWIR also have a linear correlation (Fig. 11b), albeit SWIR greatly underestimates its abundance. These results also reveal that SWIR is highly sensitive to non-mineral components such as manmade and plant materials (Figs. 6c, 7a, and 7b). In Fig. 10, asphalt is one of the most common constituents detected by SWIR and it substantially contributes to the total abundances for many samples. XRD detected actinolite in a few samples, with varied level of abundance; however, the SWIR models did not use this mineral even though it was included in the endmember bundle. Possible reasons for the discrepancies in the results obtained from XRD and SWIR are discussed next.



## 4.1 Nature of Techniques

X-ray diffraction (XRD) is the most frequent technique used to characterize dust mineralogy, nevertheless, it is less effective at detecting weakly crystalline or amorphous phases. Given that S-Q mineral abundances tend to underpredict clay mineral abundances, when the sum of all phases in the mixture is normalized to 100 percent, the abundance value for calcite and other crystalline minerals may then be overestimated. SWIR spectroscopy, being sensitive to molecular bonding rather than crystallography, provides additional information. In SWIR, clay minerals have unique features and strong absorptions, hence their abundances can be best estimated using this wavelength range. Our result determined that XRD underpredicts montmorillonite and illite abundances compared to SWIR. Therefore, we recommend using SWIR in combination with XRD for identifying and quantifying mineral dust particles, as the latter traditional approach may overlook some clay phases in the sample.

## 4.2 Limitation of Library and Modelling for Fine Grains

Natural samples have a range of particle sizes, and the minerals in the library used for modelling should match the particle size of the sample. Variable size classes (clay, silt, and sand) were present in our dust mixtures, which substantially altered the strengths of absorption features (Fig. 3) and the overall brightness of the reflectance in each sample spectrum (Gaffey, 1986; Cooper and Mustard, 1999). Gaffey (1986) showed that calcite absorption feature depth is weakened with decreasing particle size. The well characterized suite of minerals used in the USGS spectral library (Kokaly et al., 2017) often contains minerals at smaller grain sizes, but for the most part, published data use a grain size of 74-250 μm.  This larger particle size results in a relatively high spectral contrast for the library minerals. We used Hapke's equation to convert reflectance spectra to single scattering albedo (SSA). The model was able to fit the absorption features in most cases. However, as a result of the different particle sizes encountered in our samples and the library, our model used a neutral endmember (microcline) to reduce spectral contrast and match absorption feature strength of the samples (Hamilton and Christensen, 2000, and Fig. 9).

We explored whether sample particle size distribution had an effect on the quality of the model fit, particularly for the fraction of particle size greater than 30 μm. We found no systematic relationship between the quality of the model fit and the fraction of particles larger than 30 μm in the samples. Samples S15 and S1 (Fig. 6), respectively have 14 % and 33 % of their particle sizes larger than 30 μm, however, S15, with a higher fraction of fine particles, has a better modelled fit. Although we found no link between particle size and fit quality, there may still be some uncertainty in the derived abundances. Hapke models were initially derived for grain sizes larger than the wavelength, allowing geometric optics assumptions to be utilized. Many models did not match the measured spectra and so did not produced accurate mineral abundances. As a result, we recommend constructing a suite of endmembers for LSM from a spectral library that is within the same size range as typical natural dust



370    samples. This will help to reduce differences in absorption band intensities across the spectrum, which should lead to improved model fits and more accurate mineral abundances.

**4.4 Contribution of Non-Mineral Constituents**

Inspection of all model fits identified 18 samples that had poor matches (e.g., Fig. 6c). These samples showed a strong contribution of known and unknown manmade and plant materials (Figs. 7a and 7b) in their measured spectra. Among the 375    possible additional materials are a variety of particles such as asphalt, tar, styrofoam, plastic, and dry grass, some of which were visually identified. Absorption from these materials can contribute strongly to the measured spectra prohibiting a good match. Additionally, many studies have demonstrated that mixing dark grains (such as asphalt or tar) with other minerals can diminish the mixture's reflectance and considerably weaken the absorption bands observed (Nash and Conel, 1974; Singer, 1981; Clark, 1983). Calcite has a strong diagnostic absorption feature around 2340 nm, but this appears only weakly in our 380    measurements (e.g., Figs. 3, 6). The absence of this feature may be due not only to fine grain size, but also to the contribution of strong absorption from dark manmade constituents. This also leads to the underestimation of calcite abundance obtained from SWIR. XRD is not sensitive to non-crystalline phases, and thus is not sensitive to their presence in the samples. Therefore, it is preferable to use XRD to obtain abundances for crystalline phases when mixed with other materials. To characterize and quantify urban dust, reflectance spectroscopy should also be utilized to account for non-mineral materials that are present in 385    mixtures as XRD would miss them. As Fig. 7 displays, SWIR can quickly identify non-mineral diagnostic absorptions (such a hydrocarbon bonds). These materials can contribute strongly to dust mixtures collected from urban settings. Including various urban materials in spectral libraries would probably help improve the model fit but this was not in the scope of this research.

XRD detected both actinolite and kaolinite in trace and minor levels. In SWIR, actinolite was not included and kaolinite was 390    identified in only two samples. The absence of these minerals could be due to their absorption features being suppressed when mixing with other dark grains. In addition to the effect of non-mineral components, kaolinite absorption features can be weakened or disappear as montmorillonite abundances increase in the mixture (e.g., Ducasse et al., 2020).

**4.5 Obtaining Abundances from Mid and Long-wave Infrared (MWIR-LWIR)**

In VNIR/SWIR, reflectance spectra are shaped by electronic and vibrational transitions (Hunt. 1977) allowing detection of 395    compositional information of surface materials. Clay minerals commonly display sharp and narrow diagnostic absorption bands in this wavelength range (Fig. 3) and thus can be best identified and abundances estimated. For other minerals, the vibrational absorptions detectable in VNIR/SWIR are weaker signals compared to corresponding features in the mid-wave and long-wave infrared (MWIR-LWIR, ~ 2.5 to 25 µm). In particular, carbonates and silicates have very strong vibrational absorptions in MWIR-LWIR and are readily detectable in this wavelength range (e.g., Salisbury and Walter, 1989). As noted





above, SWIR is not sensitive to the common dust minerals quartz and feldspars (albite and microcline). MWIR-LWIR water absorptions in clay minerals remain strong when mixed with dark grains (Clark, 1983). Therefore, employing MWIR-LWIR may better estimate abundances of minerals that are either featureless or are obscured in VNIR/SWIR. Additionally, LWIR mineral absorption features in a mixture combine linearly (e.g., Thomson and Salisbury, 1993) allowing interpretation of measured spectra as a linear combination of its components' abundances. In future work, to identify all clays as well as quartz

and feldspars, using the full spectral range is recommended, which should identify all minerals present in the samples.

## 5 Conclusions

In this research, we set out to test if VNIR/SWIR reflectance spectroscopy combined with a Hapke model and linear spectral mixing of SSA can accurately estimate mineral abundance consistent with semi-quantitative values determined by XRD. The techniques showed general agreement after normalizing for the use of transparent minerals to match weak features in the

measured spectra. Both total clay content and carbonate are linearly correlated between the two techniques. However, XRD underpredicted total clay content and SWIR significantly underpredicted carbonate content. Our analysis showed that SWIR is well-suited to identify clay phases that would be missed by XRD techniques and is also a quick and effective way to survey a group of samples with little preparation. Figure 11a shows that spectrally derived clay abundances correlate well with XRD derived abundances, but the latter technique underpredicts clay abundances unless samples undergo time consuming additional

sample preparation. From the evaluation of SWIR spectra of dust samples, we conclude that calcite dominant absorption features are weakened when mixtures are composed of very fine-grained minerals combined with dark manmade materials. This limitation consequently leads to underprediction of calcite in the SWIR abundance determinations. SWIR is advantageous in detecting absorption features attributed to non-mineral materials in samples. These materials are common in urban settings and may also be important for radiative forcing in the atmosphere. Optical microscope images confirm the presence of black

and angular-shape materials but their composition is not readily identified with this technique. XRD, on the other hand, is not sensitive to non-crystalline phases, so it does not have the ability to detect them. While each of these approaches are useful for estimating abundances of different types of particles, a combination of the two for full characterization of urban dust has yielded good results. Based on our analysis, we recommend including spectral measurements in both MWIR-LWIR and SWIR for future studies.


Because our analysis uses VNIR/SWIR and contributes to fundamental measurements of dust, it can guide further dust mineralogy investigations by satellite imaging spectrometers such as The Earth Surface Mineral Dust Source Investigation (EMIT) (Green et al., 2020). VNIR/SWIR reflectance spectroscopy can readily identify clays, carbonates, and iron oxides, and distinguish them from non-mineral materials that are components of dust mixtures.



**Appendix A**

**Table A 1. Locality of 13 deposition samplers in Ilam city. Sample numbers shown with N/A, did not have enough samples for our analysis.**

| Samples Number | Sampling time | Latitude | Longitude | Elev. MASL |
|---|---|---|---|---|
| S1 | December 21st, 2011 (Fall) | | | |
| S2 | March 19th, 2012 (Winter) | 33°38'5.97"N | 46°24'38.26"E | 1388 |
| S3 | June 20th, 2012 (Spring) | | | |
| S4 | December 21st, 2011 (Fall) | | | |
| N/A | ~~March 19th, 2012 (Winter)~~ | 33°37'49.62"N | 46°25'27.98"E | 1404 |
| S5 | June 20th, 2012 (Spring) | | | |
| S6 | December 21st, 2011 (Fall) | | | |
| S7 | March 19th, 2012 (Winter) | 33°38'8.93"N | 46°24'46.49"E | 1400 |
| S8 | June 20th, 2012 (Spring) | | | |
| S9 | December 21st, 2011 (Fall) | | | |
| S10 | March 19th, 2012 (Winter) | 33°37'27.47"N | 46°22'22.57"E | 1295 |
| S11 | June 20th, 2012 (Spring) | | | |
| S12 | December 21st, 2011 (Fall) | | | |
| S13 | March 19th, 2012 (Winter) | 33°36'3.79"N | 46°25'13.01"E | 1438 |
| S14 | June 20th, 2012 (Spring) | | | |
| N/A | ~~December 21st, 2011 (Fall)~~ | | | |
| S15 | March 19th, 2012 (Winter) | 33°38'35.23"N | 46°24'54.96"E | 1429 |
| S16 | June 20th, 2012 (Spring) | | | |
| S17 | December 21st, 2011 (Fall) | | | |
| S18 | March 19th, 2012 (Winter) | 33°37'34.57"N | 46°25'15.31"E | 1296 |
| S19 | June 20th, 2012 (Spring) | | | |
| S20 | December 21st, 2011 (Fall) | | | |
| S21 | March 19th, 2012 (Winter) | 33°38'29.05"N | 46°24'47.64"E | 1423 |
| S22 | June 20th, 2012 (Spring) | | | |
| S23 | December 21st, 2011 (Fall) | | | |
| S24 | March 19th, 2012 (Winter) | 33°38'19.72"N | 46°26'24.21"E | 1429 |
| S25 | June 20th, 2012 (Spring) | | | |
| S26 | December 21st, 2011 (Fall) | | | |
| S27 | March 19th, 2012 (Winter) | 33°37'26.04"N | 46°24'46.38"E | 1376 |
| S28 | June 20th, 2012 (Spring) | | | |
| S29 | December 21st, 2011 (Fall) | | | |
| S30 | March 19th, 2012 (Winter) | 33°38'42.44"N | 46°24'57.64"E | 1462 |
| S31 | June 20th, 2012 (Spring) | | | |
| S32 | December 21st, 2011 (Fall) | | | |
| S33 | March 19th, 2012 (Winter) | 33°38'21.68"N | 46°23'56.16"E | 1395 |
| S34 | June 20th, 2012 (Spring) | | | |
| S35 | December 21st, 2011 (Fall) | | | |
| S36 | March 19th, 2012 (Winter) | 33°38'1.49"N | 46°23'58.75"E | 1370 |
| S37 | June 20th, 2012 (Spring) | | | |





**Appendix B**

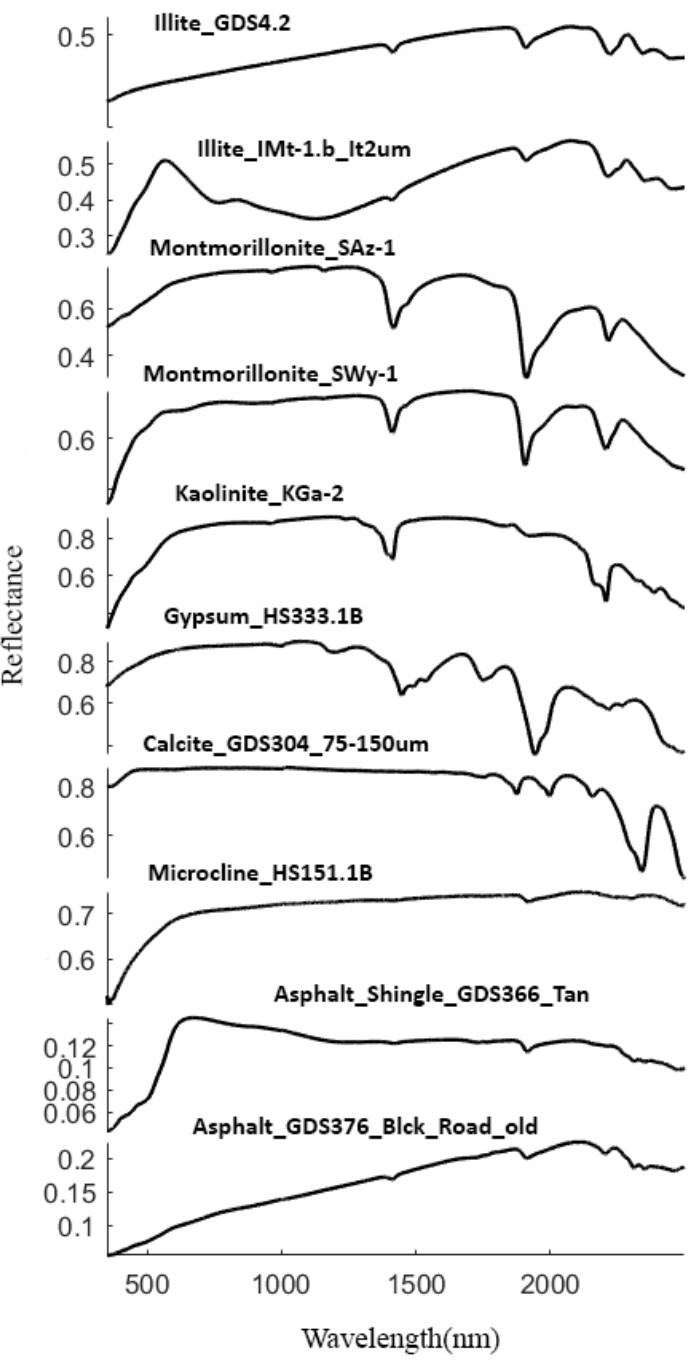

**Figure B 1. Mineral spectra from USGS library (Kokaly et al., 2017) used by us to retrieve mineral abundances for natural dust samples.**

*Data and code availability*. Data and code used in this study are available on request to msadrian@nevada.unr.edu or wcalvin@unr.edu.

*Author contributions.* MRS and WMC collaborated on project conceptualization, funding and goals. MRS performed all measurements and data analysis. JM provided OM image interpretation. MRS prepared the manuscript with contributions from all co-authors.

*Competing interests.* The authors declare that they have no conflict of interest.

### Acknowledgment

This work has been supported in part by the UNR Graduate Student Association Graduate Research Grant and Travel Grant Programs, the College of Science Dean's Office, Nevada NASA EPSCoR Research Infrastructure Seed Grant #18-83 from Federal Award number NNX15AK48a, and coauthor (WMC) discretionary funds. The authors thank UNR Chemistry Department Shared Instrumentation Laboratory for making its XRD facilities available, Janina Ruprecht for assisting with XRD training, Mohammad Jafari who helped with writing algorithms in Matlab, and Patrick Arnott and Hans Moosmuller for

their helpful comments on the draft manuscript.

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
