# Peer review of "Contrasting Mineral Dust Abundances from X-Ray Diffraction and Reflectance Spectroscopy"

_Atmospheric Measurement Techniques, 2021_

## Referee Comment (RC2)

[referee-annotated manuscript omitted]

---

## Author Comment (AC1)

**Responses to reviewer #1**

We thank the anonymous reviewer for their valuable and constructive comments/suggestions on our manuscript. We have revised the manuscript accordingly and please find our point-by-point responses below.

1. **This is a methodology study and should have a standard proportionally mixed samples for comparison. So, the results for the mineral percentages based on combined XRD and IR methods may still not convincible. At least, the calculation results have to be tested by the standard mixture.**

**Response:** We understand the request, but it is impractical to have a standard proportionally mixed dust samples which include several natural dust-forming minerals for the purpose of XRD and IR comparison. XRD has been proven to be one of the most important analytical approaches used in the qualitative and semi-quantitative study of crystalline samples, and the calculation results for IR has been tested in many past studies and shown to be in a good agreement with the standard (physical) mixture of minerals. Additional details are provided here and we have included additional clarification in the revised manuscript.

As it is already noted in the manuscript, XRD has been used as a common, reliable technique to obtain minerology and semi-quantitative phase abundances for atmospheric dust samples (e.g., Caquineau et al., 1997; Nowak et al., 2018). Also, many studies have employed radiative transfer (RT) to model reflectance spectra of synthetic or laboratory mineral mixtures, validating the derived abundances. For example, Robertson et al., (2016) demonstrated that physical mixtures of clay and sulfate at varying abundances were accurately determined (within 5 %) using a Hapke RT model. Another study from Poulet and Erard (2004) showed that the mineral abundances derived using nonlinear spectral mixing are accurate to within 5-10 % for the laboratory mixture of pyroxenes, olivine, plagioclase, with various particle sizes.

Moreover, there is a large body of research that show the mineral abundances derived from XRD are in good agreement with IR reflectance spectra derived abundances (e.g., Thorpe et al., 2015; Pan et al., 2015; Leask and Ehlmann, 2016). For example, Leask and Ehlmann (2016) performed measurements on 15 rock samples (with various particle size) collected from Oman, and they found that VSWIR reflectance spectroscopy paired with linear spectral unmixing yields quantitative mineral abundance estimates that are consistent (within 10-15 %) with XRD abundance estimations.

These prior studies have validated the approach for mixtures of a few minerals and the goal of our study is to test this on natural dust samples that are a complex mixture of many natural minerals. The number of mineral phases in these samples varies from six to nine individual minerals, therefore, it is nearly impossible (and is impractical) to create complex physical models in widely varying abundances as observed in our samples for the purpose of comparison.

For additional clarification we added the following sentence to the introduction section of the manuscript's text, after the line which explains about Robertson et al., (2016) testing the accuracy of RT models through comparisons with physical mixtures.

"Multiple past studies have shown that the mineral abundances (for rocks and rocking forming fine grained mineral samples) derived from visible and infrared reflectance spectra are in good agreement with minerals abundances that are obtained using XRD (e.g., Pan et al., 2015; Thorpe et al., 2015; Leask and Ehlmann, 2016). For example, Leask and Ehlmann (2016) performed measurements on 15 rock samples (with various particle sizes) collected from Oman, and they found that VSWIR reflectance spectroscopy paired with linear spectral unmixing yields quantitative mineral abundance estimates that are consistent (within 10-15 %) with XRD abundance estimations."

References (* represents the new references that have been added to the manuscript)

Caquineau, S., Magonthier, M. C., Gaudichet, A., and Gomes, L.: An improved procedure for the X-ray diffraction analysis of low-mass atmospheric dust samples, European Journal of Mineralogy, 9, 157-166, 1997.

*Leask, E. K., Ehlmann, B. L., and Ieee: Identifying and Quantifying Mineral Abundance Through VSWIR Microimaging Spectroscopy: A Comparison to XRD and SEM, 2016 8th Workshop on Hyperspectral Image and Signal Processing: Evolution in Remote Sensing (Whispers), 5, 2016.

Nowak, S., Lafon, S., Caquineau, S., Journet, E., and Laurent, B.: Quantitative study of the mineralogical composition of mineral dust aerosols by X-ray diffraction, Talanta, 186, 133-139, 10.1016/j.talanta.2018.03.059, 2018.

*Pan, C., Rogers, A. D., and Thorpe, M. T.: Quantitative compositional analysis of sedimentary materials using thermal emission spectroscopy: 2. Application to compacted fine-grained mineral mixtures and assessment of applicability of partial least squares methods, Journal of Geophysical Research-Planets, 120, 1984-2001, 10.1002/2015je004881, 2015.

Poulet, F. and Erard, S.: Nonlinear spectral mixing: Quantitative analysis of laboratory mineral mixtures, Journal of Geophysical Research-Planets, 109, 12, 10.1029/2003je002179, 2004.

Robertson, K. M., Milliken, R. E., and Li, S.: Estimating mineral abundances of clay and gypsum mixtures using radiative transfer models applied to visible-near infrared reflectance spectra, Icarus, 277, 171-186, 10.1016/j.icarus.2016.04.034, 2016.

*Thorpe, M. T., Rogers, A. D., Bristow, T. F., and Pan, C.: Quantitative compositional analysis of sedimentary materials using thermal emission spectroscopy: 1. Application to sedimentary rocks, Journal of Geophysical Research-Planets, 120, 1956-1983, 10.1002/2015je004863, 2015.

2. **The calculation of the mineral contents based on the XRD patterns of the natural samples should be careful, as the peaks may be the overlapped results of several different minerals, such as the peak for the kaolinite might contain d(002) peak of chlorite, etc.**

**Response:** Without special sample preparation XRD might miss detecting some clay minerals, whereas VSWIR spectroscopy is a complementary technique used both to obtain new phases (such as amorphous phases or partly crystalline clay minerals that are probably not detected by XRD) and to confirm the phases that are already identified by XRD. Chlorite has diagnostic absorption features in VSWIR, due to several iron absorption bands between 400 to 1100 nm, and characteristic complex doublet absorption near 2330 and 2390 nm, which makes it readily identifiable in this spectral range (VSWIR) (e.g., King and Clark, 1989; Kokaly et al., 2017). We thoroughly inspected all of the samples' IR spectra (specifically for clays such as chlorite and various iron oxides) and we could not find spectral signature attributed to chlorite in these samples. Therefore we concluded the peak for kaolinite does not contain d(002) peak of chlorite, and is solely attributed to kaolinite.

References

Clark, R. N., King, T. V. V., Klejwa, M., Swayze, G. A., and Vergo, N.: High Spectral Resolution Reflectance Spectroscopy of Minerals, Journal of Geophysical Research-Solid Earth and Planets, 95, 12653-12680, 10.1029/JB095iB08p12653, 1990.

King, T. V. V. and Clark, R. N.: Spectral Characteristics of Chlorites and mg-Serpentines Using High-Resolution Reflectance Spectroscopy, Journal of Geophysical Research-Solid Earth and Planets, 94, 13997-14008, 10.1029/JB094iB10p13997, 1989.

Kokaly, R. F., Clark, R. N., Swayze, G. A., Livo, K. E., Hoefen, T. M., Pearson, N. C., Wise, R. A., Benzel, W. M., Lowers, H. A., Driscoll, R. L., and Klein, A. J.: USGS Spectral Library Version 7, Reston, VA, Report 1035, 68, 10.3133/ds1035, 2017.

3. **Should the relative proportion of different compositional groups be reported as volume percentage or area percentage or weight percentage? If it the weight percentage, some more conversion factors must be used according to the world standard such as American Standard for the XRD-based mineral identification?**

**Response:** XRD semi-quantitative (S-Q) abundance analysis is calibrated by mass, and calculates the weight percentage (wt. %) of the phases present in the samples. As describe in Sect 2.6, the output result for the spectral modelling is mass fractional abundance for each component in the sample. The mass fraction is also known as the

mass percentage (mass %) and is equivalent to the weight percentage. We added wt. % to the caption of figure 8 to clarify the units for XRD S-Q measurements.

4. **Line 104-106: It is stated "Although there were no standard reference patterns in the AMCSD dataset to show the match peak for illite, we detected this mineral in S11 based on the visual assessment and past published data on the location of illite peaks". Does it mean that the "illite" mineral in the analyzed samples are the illite/smectite mixed layer with the d(001) from 10 to 15 Å, i.e., the transition between the illite and smectite?**

**Response:** Our intent was not clearly conveyed as both reviewers misunderstood this discussion.

While using the DIFFRAC.EVA, software to perform the XRD evaluation, we compared the dust sample patterns to a reference database of minerals to identify peaks in the sample. The illite reference pattern is included in the software database and this mineral was readily and confidently detected. However, we were unable to export the reference illite data (e.g., text or XY format files) from the DIFFRAC.EVA software in order to plot it in Figure 2 and demonstrate it as a standard reference. Therefore, we used two highly-cited published studies (Gualtieri (2000) and Drits et al., (2010)) as references for the location of the illite peak.

We have revised the text to read:

"The identification of the illite peak in Fig 2 uses data from the published literature such as from Gualtieri (2000) and Drits et al., (2010). While this peak pattern was available in the DIFFRAC.EVA software we were not able to export the reference patterns in order to show them in Fig 2. We used the AMCSD database for other minerals shown in Fig 2, but this database does not include a pattern for illite."

References

Drits, V. A., Zviagina, B. B., McCarty, D. K., and Salyn, A. L.: Factors responsible for crystal-chemical variations in the solid solutions from illite to aluminoceladonite and from glauconite to celadonite, American Mineralogist, 95, 348-361, 10.2138/am.2010.3300, 2010.

Gualtieri, A. F.: Accuracy of XRPD QPA using the combined Rietveld-RIR method, Journal of Applied Crystallography, 33, 267-278, 10.1107/s002188989901643x, 2000.

**5. The percentages of detected minerals measured by XRD are different from measured by SWIR. Compared with SWIR, the XRD overestimate the quartz and calcite contents, and underestimate the clay mineral contents. However, which results can we trust, XRD or SWIR?**

**Response:** Past studies reported a detection limit of generally < 2 % for well crystalline minerals and an uncertainty of approximately ±10 % related to XRD mineral quantification. (e.g., Bish and Chipera, 1991)." Our analysis showed that SWIR reflectance spectroscopy is well-suited to identify and quantify clay phases (such as Montmorillonite) that would be missed by XRD techniques and is also a quick and effective way to survey a group of samples with little preparation. In addition, SWIR is advantageous in detecting absorption features attributed to non-mineral materials in samples (such as asphalt), as well as amorphous and partly crystalline minerals. XRD, on the other hand, can reliably identify well crystalline minerals, nevertheless, it is less effective at detecting weakly crystalline (some clays) or amorphous phases (such as manmade materials). Further, XRD can accurately detect quartz, feldspars, calcite, and amphibole in dust samples, and estimates relatively reliable abundances for them. However, SWIR could not identify these silicates and carbonates due to either not being absorbed by them or the limitations incurred by fine grain size or dark manmade materials that are present in the sample. Our results indicate that it is beneficial to use both XRD and reflectance spectroscopy to characterize natural airfall dust, because the former technique is good at identifying and quantifying the SWIR-transparent minerals (e.g., quartz, albite, and microcline), while the latter technique is superior for determining abundances for clays and non-mineral components.

Regarding reviewer's comment, we have added the following statement to the conclusion section.

"Because quartz and feldspars are substantial fractions of total mineral abundances of dust samples (Fig. 8), we suggest the use of XRD as an initial reliable method for mineral identification and quantification. Based on our analysis, we recommend that future research include spectral measurements in both VSWIR and LWIR, as the latter spectral range can be complementary to the former and obtained abundances for SWIR-transparent minerals (e.g., quartz and feldspars). As a result, the present minerals in the bulk sample can qualitatively and quantitatively assessed by both VSWIR and LWIR, and then confidently compared with XRD determined mineral abundances."

Reference (* represents the new reference that has been added to the manuscript)

*Bish, D. L. and Chipera, S. J.: Detection of Trace Amounts of Erionite Using X-Ray-Powder Diffraction - Erionite In Tuffs of Yucca Mountain, Nevada, And Central Turkey, Clays and Clay Minerals, 39, 437-445, 10.1346/ccmn.1991.0390413, 1991.

6. **Line 166: "The modes for each size ranges are clay ~ 6 %, silt ~ 85 %, and sand ~ 3 %." From Figure 5, it looks that the "Sand ~ 3 %" is not correct.**

**Response:** The mode for sand was double-checked and the reviewer is correct. In order to avoid confusion, we have changed to the mean values for each size range and show these with a new annotation on the ternary diagram.

7. **The section title sequence number needs to be revised.**

**Response:** The section title sequence number has been revised.

---

## Author Comment (AC2)

**Response to reviewer #2**

We thank the reviewer for his insightful and constructive comments/suggestions, and revisions on our manuscript. Wording and grammatical changes were adapted and included in the revised version of manuscript. More substantive comments are addressed point-by-point below.

1. **The experiment is well designed but could benefit by testing the spectroscopic model with a few constructed mineral mixtures with known weight percent constituents. Seems like some lab prepared mixtures of these mineral components and asphalt with known weight percents could be used to test the overall accuracy of the SSD spectral model. They could also be used to test the sensitivity of XRD to clay species. This seem like a necessary step to demonstrate the effectiveness of the spectral model.**

**Response:**

We understand the request, but it is impractical to have a standard proportionally mixed dust samples which include several natural dust-forming minerals for the purpose of XRD and IR comparison. XRD has been proven to be one of the most important analytical approaches used in the qualitative and semi-quantitative study of crystalline samples, and the calculation results for IR has been tested in many past studies and shown to be in a good agreement with the standard (physical) mixture of minerals. Additional details are provided here and we have included additional clarification in the revised manuscript.

As it is already noted in the paper's text, XRD has been used as a common, reliable technique to obtain mineralogy and semi-quantitative phase abundances for atmospheric dust samples (e.g., Caquineau et al., 1997; Nowak et al., 2018). Also, many studies have employed radiative transfer (RT) to model reflectance spectra of synthetic or laboratory mineral mixtures, validating the derived abundances. For example, Robertson et al., (2016) demonstrated that physical mixtures of clay and sulfate at varying abundances were accurately determined (within 5 %) using a Hapke RT model. Another study from Poulet and Erard (2004) showed that the mineral abundances derived using nonlinear spectral mixing are accurate to within 5-10 % for the laboratory mixture of pyroxenes, olivine, plagioclase, with various particle sizes.

Moreover, there is a large body of research that show the mineral abundances derived from XRD are in good agreement with IR reflectance spectra derived abundances (e.g., Thorpe et al., 2015; Pan et al., 2015; Leask and Ehlmann, 2016). For example, Leask and Ehlmann (2016) performed measurements on 15 rock samples (with various particle size) collected from Oman, and they found that VSWIR reflectance spectroscopy paired with linear spectral unmixing yields quantitative mineral abundance estimates that are consistent (within 10-15 %) with XRD abundance estimations.

These prior studies have validated the approach for mixtures of a few minerals and the goal of our study is to test this on natural dust samples that are a complex mixture of many natural minerals. The number of mineral phases in these

samples varies from six to nine individual minerals, therefore, it is nearly impossible (and is impractical) to create complex physical models in widely varying abundances as observed in our samples for the purpose of comparison.

For additional clarification we added the following sentence to the introduction section of the manuscript's text, after the line which explains about Robertson et al., (2016) testing the accuracy of RT models through comparisons with physical mixtures.

"Multiple past studies have shown that the mineral abundances (for rocks and rocking forming fine grained mineral samples) derived from visible and infrared reflectance spectra are in good agreement with minerals abundances that are obtained using XRD (e.g., Pan et al., 2015; Thorpe et al., 2015; Leask and Ehlmann, 2016). For example, Leask and Ehlmann (2016) performed measurements on 15 rock samples (with various particle size) collected from Oman, and they found that VSWIR reflectance spectroscopy paired with linear spectral unmixing yields quantitative mineral abundance estimates that are consistent (within 10-15 %) with XRD abundance estimations."

References (* represents the new references that have been added to the manuscript)

Caquineau, S., Magonthier, M. C., Gaudichet, A., and Gomes, L.: An improved procedure for the X-ray diffraction analysis of low-mass atmospheric dust samples, European Journal of Mineralogy, 9, 157-166, 1997.

*Leask, E. K., Ehlmann, B. L., and Ieee: Identifying and Quantifying Mineral Abundance Through VSWIR Microimaging Spectroscopy: A Comparison to XRD and SEM, 2016 8th Workshop on Hyperspectral Image and Signal Processing: Evolution in Remote Sensing (Whispers), 5, 2016.

Nowak, S., Lafon, S., Caquineau, S., Journet, E., and Laurent, B.: Quantitative study of the mineralogical composition of mineral dust aerosols by X-ray diffraction, Talanta, 186, 133-139, 10.1016/j.talanta.2018.03.059, 2018.

*Pan, C., Rogers, A. D., and Thorpe, M. T.: Quantitative compositional analysis of sedimentary materials using thermal emission spectroscopy: 2. Application to compacted fine-grained mineral mixtures and assessment of applicability of partial least squares methods, Journal of Geophysical Research-Planets, 120, 1984-2001, 10.1002/2015je004881, 2015.

Poulet, F. and Erard, S.: Nonlinear spectral mixing: Quantitative analysis of laboratory mineral mixtures, Journal of Geophysical Research-Planets, 109, 12, 10.1029/2003je002179, 2004.

Robertson, K. M., Milliken, R. E., and Li, S.: Estimating mineral abundances of clay and gypsum mixtures using radiative transfer models applied to visible-near infrared reflectance spectra, Icarus, 277, 171-186, 10.1016/j.icarus.2016.04.034, 2016.

*Thorpe, M. T., Rogers, A. D., Bristow, T. F., and Pan, C.: Quantitative compositional analysis of sedimentary materials using thermal emission spectroscopy: 1. Application to sedimentary rocks, Journal of Geophysical Research-Planets, 120, 1956-1983, 10.1002/2015je004863, 2015.

2. **Amorphous "humps" in XRD patterns can be used to qualitatively assess samples for the presence of non-crystalline phases and they should be re-examined for this evidence.**

**Response:** We thoroughly examined all samples' XRD plots, and we do not observe any discernible "humps" or broad features that could be attributed to non-crystalline phases.

3. **Modifying the statement "To date, VNIR/SWIR spectroscopy has not been used to study natural dust particle mineralogy; however, it can provide quantitative measurements and identify both amorphous and crystalline phases (Clark, 1999)".**

**Response:** We modified the statement to "To date, very limited studies have used VSWIR to determine natural dust particle mineralogy (e.g., Reynolds et al., 2020); however, it can provide quantitative measurements and identify both amorphous and crystalline phases (Clark, 1999)".

We added the following reference from Reynolds et al., (2020), to the reference section.

Reynolds, R. L., Goldstein, H. L., Moskowitz, B. M., Kokaly, R. F., Munson, S. M., Solheid, P., Breit, G. N., Lawrence, C. R., and Derry, J.: Dust Deposited on Snow Cover in the San Juan Mountains, Colorado, 2011-2016: Compositional Variability Bearing on Snow-Melt Effects, Journal of Geophysical Research-Atmospheres, 125, 24, 10.1029/2019jd032210, 2020.

4. **Map needs a scale in km. Are the lats and longs for the image corners?**
**Response:** Scale in km is added to the map in the revised manuscript. The statement "Shown latitudes and longitudes are the coordinates for the corners of the map", was added to the caption of Figure 1.

5. **Why not use the "JADE" database? Illite seems like a pretty common mineral. Explain.**

**Response:** Our intent was not clearly conveyed as both reviewers misunderstood this discussion.

While using the DIFFRAC.EVA, software to perform the XRD evaluation, we compared the dust sample patterns to a reference database of minerals to identify peaks in the sample. The illite reference pattern is included in the software database and this mineral was readily and confidently detected. However, we were unable to export the reference illite data (e.g., text or XY format files) from the DIFFRAC.EVA software in order to plot it in Figure

2 and demonstrate it as a standard reference. Therefore, we used two highly-cited published studies (Gualtieri (2000) and Drits et al., (2010)) as references for the location of the illite peak.

We have revised the text to read:

"The identification of the illite peak in Fig 2 uses data from the published literature such as from Gualtieri (2000) and Drits et al., (2010). While this peak pattern was available in the DIFFRAC.EVA software we were not able to export the reference patterns in order to show them in Fig 2. We used the AMCSD database for other minerals shown in Fig 2, but this database does not include a pattern for illite."

6. **Why weren't clay separations done?  Explain.**

**Response:** The following statement was added to the Sect. 2.2 X-Ray Diffraction (XRD) to clarify why clay separation were not done:

"Because the volumes of dust samples were low, XRD sample preparation specifically for clay minerals was not conducted. Also, we could not follow sample preparation developed for low mass atmospheric dust samples (Caquineau et al., 1997) due to a lack of access to specialized equipment."

7. **Add a few sentences that discuss the lower thresholds for semi-quant XRD.  Can it detect kaolinite at less than 5wt%?  This might help explain differences between what spectroscopy and XRD detect.**

**Response:** The following statement was added to Sect. 2.2 X-Ray Diffraction (XRD) to clarify the XRD detection limit:

"S-Q abundances made from the diffraction measurements are derived from relative proportion of minerals (weight percentage %) in the sample that should add up to 100 %. Given that the XRD is less effective at detecting and quantifying poorly crystalline minerals and amorphous phases (Moore and Reynolds, 1997), the obtained abundance results for other existing well crystalline minerals can be overestimated. Past studies reported a detection limit of generally < 2 % for well crystalline minerals and an uncertainty of approximately ±10 % related to XRD mineral quantification. (e.g., Bish and Chipera, 1991)."

We added the following reference from Bish and Chipera, (1991), to the reference section.

Bish, D. L. and Chipera, S. J.: Detection of Trace Amounts of Erionite Using X-Ray-Powder Diffraction - Erionite In Tuffs of Yucca Mountain, Nevada, And Central Turkey, Clays and Clay Minerals, 39, 437-445, 10.1346/ccmn.1991.0390413, 1991.

8. **In the figure caption state that illite (I) was not in the XRD database.**

**Response:** Please refer to the response to point #5 above.

9. **I'm impress if you have enough halon around to press your own calibration standard. More likely this is a Spectralon calibration plate.**

**Response:** The reviewer is correct and we replaced the word "halon" with "Spectralon".

10. **Where the spectrum further corrected to absolute reflectance to remove the weak spectral signature of the spectralon 2.1 μm feature? If not explain why not?**

**Response:** Only figure 3 is presented as relative reflectance. Both Figure 6 and 7 are single scattering albedo (SSA). The reviewer correctly pointed out that we should update the relative reflectance to absolute reflectance. Following Kokaly et al., (2017) (or https://crustal.usgs.gov/speclab/data/HTMLmetadata/README/content.htm),

$R(\lambda) = R_{rel}(\lambda) * R_{Spectralon}(\lambda)$, (Eq.1)

(Where $R(\lambda)$, $R_{rel}(\lambda)$, and $R_{Spectralon}(\lambda)$ are absolute, relative and Spectralon reflectance), we converted samples' spectral measurement ($R_{rel}(\lambda)$) to absolute reflectance ($R(\lambda)$). The final manuscript will include updated figures. While further correcting the spectrum to absolute reflectance yielded better modelled fit for some samples, it had no significant influence on the final mineral abundances or our conclusions.

11. **There is no spectral evidence of illite in either of these three sample spectra. Sample S15's spectrum lacks a 2.35 μm absorption that should be present if illite were present. There is no convincing spectral evidence of illite in this sample based on its VSWIR spectrum.**

**Response:** In order to show the unique illite absorption feature near 2345 nm we are including a different sample in Figure 2 where this feature is more obvious. This will be provided in the revised manuscript.

12. **poorly stated reason. More like we show only that portion of the spectra between 1350 and 2500 microns to concentrate etc.**

**Response:** We replaced the statement "We truncated all spectral plots at 1350 nm in order to focus on the above 1350 nm spectral range with the strongest features. These samples do not include iron oxides, therefore exclusion of the spectral range from 350 to 1350 nm will not miss any major mineral components." with the following statement for the purpose of clarification:

"Since we did not see absorption features attributed to iron oxides in these samples, we truncated all spectral plots at 1350 nm in order to focus on spectral range above 1350 nm with the strongest features. Therefore, exclusion of the spectral range from 350 to 1350 nm will not miss any major mineral components."

13. **How was the absence of iron oxide/hydroxide mineral determined? Were they not identified by XRD? No observed absorptions in the visible? Some of the "dark" grains in Fig. 4a look like opaques usually attributed to Fe-bearing minerals if these are not grains of asphalt. Could it be that the concentration of Fe oxide/hydroxides in these samples is just below the XRD detection threshold? What is that threshold? Some discussion is needed on this topic, as a few hundred ppm hematite or**

**goethite stains rocks red or brown. Is doesn't take much of these minerals in dust but can have a profound effect on radiative forcing of the atmosphere. That is why we are mapping them with EMIT.**

Response: The color of the Ilam dust samples is gray. There are no samples with red or brown color. We thoroughly inspected all of the samples, in particular in the wavelength range between 350 to 1250 nm, and observed no spectral signature attributed to iron oxide/hydroxide minerals. XRD also showed no peaks related oxide/hydroxide minerals. The dark grain materials shown with optical microscopy (Fig. 4a) are attributed to asphalt as it was confirmed by reflectance spectroscopy (Fig. 6b, 6c, Fig. 7a, and Fig. 9).

**14. They might be glass (in figure 4, from optical microscopy).**

Response: Please see the response to point #2 above.

**15. Any supporting reference for this decision (why wet measurements were performed)?**

Response: This method provides more accurate size distributions. We added Hartshorn et al., (2021) to the manuscript, as the supporting reference of why we selected wet measurement.

We added the following reference from Hartshorn et al., (2021), to the reference section.

Hartshorn, E. J., McDonald, E. V., Weir, W. B., Sweeney, M., Houseman, S. M., Lacey, T.: An Integrated Model Combining UAS Imagery and PI-SWERL for Evaluating Intra-Landform Dust Emission Variability, Report Prepared for U.S. Army Corps of Engineers Engineer Research and Development Center Cold Regions Research and Engineering Laboratory, 2021.

**16. I'm having trouble see how the mode for sand is 3%.**

Response: The mode for sand was double-checked and the reviewer is correct. In order to avoid confusion, we have changed to the mean values for each size range and show these with a new annotation on the ternary diagram.

**17. Add a sentence or two that justifies this assumption (line 227, in initial manuscript). What part does the assumption of a uniform grain size play in the overall accuracy of the spectral modeling? Of course, this assumption may be valid given the very fine-grained nature of dust.**

Response: We revised the sentence as follows to justify our assumption.

"These spectra are shown in Appendix B (Fig. B1). Most library minerals used were in the grain size range < 150 µm. Our samples have narrow grain size range (Fig. 5) so that our model assumes all components have the same grain size and does not allow this to vary as a free parameter."

**18. The biggest discrepancy is the inverse spectralon absorption at 2.1µm. This needs to be discussed. The hump at 2.1µm is from a spectralon absorption. It would disappear if the dust spectra were corrected to absolute reflectance as the USGS spectral library spectra are.**

Response: Please see the response to point #10 above.

**19. Is this weight % or volume %. Please clarify in the (Fig. 8) caption.**

**Response:** Weight percentage (wt. %) was added to the caption for figure 8.

**20. I don't see a "good" spectral model fit to XRD for the checkmarked samples unless +/- 5wt% calcite is "good." Also, there is no hint of actinolite spectrally in some checkmarked samples. How can the match be "good" (caption for Fig. 10)?**

**Response:** The samples displayed with check marks (in Fig. 10) had a spectral fit that was relatively well modelled based on our visual evaluation and a low RMSE. For example, Fig 6a shows SSA for a good spectral fit. We assumed the samples with a good spectral fit produce relatively accurate minerals abundances and selected them for the purpose of comparison with XRD derived abundances as discussed in Section 3.2.

We revised the statement in the caption for figure 10 for the purpose for clarification:

"Figure 10. Bar charts show XRD (top) and SWIR (bottom) normalized abundances after removing the transparent minerals. Those samples with check marks had relatively well-modelled spectral fit (e.g., Figs. 6a and 6b) as described in the text, and are used for subsequent comparison as described in Sect. 3.2."

**21. An amorphous hump for the organic component of the asphalt should show in the XRD patterns. Quantifying this might be difficult unless standards with known weight percent organics where used to derive a calibration for the intensity of the hump.**

**Response:** Please see the response to comment number 2. Also, as we discussed in the manuscript we were interested in minerology and not the manmade constituents present in the samples.

**22. Certainly, dark asphalt may suppress the scattering of photons and its effect at low abundance might confound Hapke modeling when it is present even at low abundances. But the oxidized asphalt spectrum shown in fig. 7a has 70 % reflectance in the SWIR. This is pretty bright so shouldn't be spectrally dominant the way fresh unoxidized dark asphalt would.**

**Response:** The reviewer has misinterpreted Figure 7. The spectra for asphalt and tar shown are presented as single scattering albedo. They have absolute reflectance values lower than 23 %, contributing as dark agents in dust samples.

To clarify about asphalt reflectance value, we added the following statement to Section 4.3.

"It is good to note that the reflectance values for asphalt and tar in the USGS library (Kokaly et al., 2017) are less than 23 %."

**23. This is not quite true in my experience. I routinely use spectroscopy to gauge the crystallinity of kaolinite and alunite. I would say spectroscopy is sensitive to both molecular bonding and in many cases crystalline order.**

**Response:** We replaced the statement "SWIR spectroscopy, being sensitive to molecular bonding rather than crystallography, provides additional information." with:

"SWIR spectroscopy, being sensitive to molecular bonding, provides additional information."

**24. There is a spectrum of this in the USGS library labeled polystyrene.**

**Response:** As shown in the image below, the Styrofoam materials we measured (Sadrian et al., 2021) are distinct from the polystyrene spectra from USGS library (Kokaly et al., 2017).

[Figure]

**Image shows the differences between the polystyrene spectra (red line) exported from USGS library (Kokaly et al., 2017) and the Styrofoam spectra (black line) that Sadrian et al., (2021) measured in the lab.**

**25. Terms:**

We replaced "VNIR/SWIR" with "VSWIR" in whole manuscript, and "full spectral range" with "combined VSWIR and LWIR ranges" in Sect. 4.5.

---

## Author Response (AR2)

04/26/2022

Dear associate editor,
Dr. Mingjin Tang,

We have responded to all reviewer #3 comments as provided in the details in the following pages.

We hope that with these revisions and given the timeliness of the research that you will find this manuscript suitable for acceptance and publication in AMT.

Sincerely,

Mohammad Reza Sadrian

**Responses to reviewer #3**

We thank the anonymous reviewer for their valuable and constructive comments/suggestions on our manuscript. We have revised the manuscript accordingly and please find our point-by-point responses below.

**This study is well conducted and clearly reported. Measurements and data processing are robust and properly explained. The interpretations of the results of reflectance spectroscopy and XRD and their comparison are convincing, as well as the ensuing discussions. This is thus a nice methodological study. After the answers to the previous reviewers, there remains, to me, the following issues:**

**1. A part of experimental protocol is missing: it isn't described how dust is extracted from the MDCO. The way it is extracted (dry or wet for instance) could have an effect on measurements. If available, the mass of each collected sample could be added in the Table A.1, or at least orders of magnitude of the collected quantities. Sample preparation and used quantities should also be specified for each analytical method.**

**Responses:** Regarding the reviewer's point that "how dust is extracted from the MDCO", we added the following statement to the section 2.1 Sample Collection:

"*As part of sample collection procedure, first, dust samples in MDCO were dried at room temperature to preserve the mineralogical and physical properties of the surface soils from which they were transported. Then, the dry samples were collected from the samplers by thoroughly cleaning the dust depositions using a brush. All samples were transferred to separate plastic bottles for the further experiments.*"

Regarding the reviewer's point that "If available, the mass of each collected sample could be added in the Table A.1, or at least orders of magnitude of the collected quantities", we would like to mention that Sadrian et al., 2012 already reported the mass of each collected sample, and here we include the minimum and maximum mass of the collected dust to the manuscript section 2.1 Sample Collection, using the following statement:

"*The mass for the collected dust samples ranges from minimum ~ 0.01 g to maximum ~ 5 g.*"

Regarding the reviewer's point that "Sample preparation and used quantities should also be specified for each analytical method", we added the following statements to the section 2.1 Sample Collection:

"*It should be noted that there was no special sample preparation that performed for the purpose dust measurements (with XRD and spectroscopy) described in the next sections. Prior to these measurements unwanted debris as well as detectable manmade and plant materials were removed from the samples and we made sure to use similar quantity of dust (~ 1 g) for each of experiments that were conducted with spectroscopy, and for the measurements that were collected using XRD.*"

Reference

Sadrian, M. R., Mohammadkhan, S., Mashhadi, N., Alavipanah, S. K., and Dashtakian, K.: Analyzing and investigation of dustfall by MDCO (case study: the city of Ilam), International desert research center, University of Tehran, 2012.

**2. The text should be revised avoiding confusion between atmospheric dust samples and dust deposition samples. Indeed, atmospheric dust refers rather to particles in suspension in the air, which can be long-range transported, with a very fine particle size. The study is carried out here on deposition samples with coarser particle size. Since the analytical methods used are sensitive to the amount of matter, the signal is dominated by the response of coarse particles rather than fine ones. Then these potentially airborne fine particles are negligible in such measurements. The analysis of atmospheric dust also presents different analytical constraints**

**(because the samples are collected in very small masses, usually on filters). To be as explicit as possible, it would be better here to use "dust deposition".**

**Response:** In the manuscript's abstract with replaced "atmospheric dust samples" with "*dust deposition samples*" to avoid confusion and be as explicit as possible.

**3. Following the remark of previous reviewer on the presence/absence of iron oxides, I think a better evidence that these black grains are asphalt would be appreciable. The fact that iron oxides are not detected (while we know that, when they are present, they often remain below the detection limits of the XRD) and that the asphalt is detected by SWIR is a bit short argument.**

**Response:** In section 2.6 Mineral Abundance Estimation from Reflectance Spectra, we stated that "In addition to minerals, we found hydrocarbon (C-H) absorption features related to asphalt and tar in many of the samples in our preliminary analysis, and thus we included their spectra (Fig. 7a) in the input endmember bundles for modelling all 37 samples." Therefore, according to this statement and figure 7a, asphalt has a doublet diagnostic absorption feature between 2280 and 2370 nm in SWIR (which is also identified clearly in e.g., Samples 32 and 33 in Appendix C, Fig. C 1).

Regarding the presence/absence of iron oxides in the samples, in section 2.3 VSWIR Reflectance Spectroscopy of manuscript, we stated that "Since we did not see absorption features attributed to iron oxides in these samples, we truncated all spectral plots at 1350 nm in order to focus on spectral range above 1350 nm with the strongest features (SWIR range). Therefore, exclusion of the spectral range from 350 to 1350 nm will not miss any major mineral components."

We should add that iron oxides have strong absorption features in the visible and near infrared (up to around 1000 nm), however we did not detect any iron oxides in these samples' spectra. Therefore, to clarify about the presence/absence of iron oxides, we changed the above manuscript's statement to read:

*"Iron oxides have strong diagnostic spectral signatures in the visible and near infrared (up to around 1000 nm), however we did not see absorption features attributed to them in these samples. Therefore, we truncated all spectral plots at 1350 nm in order to focus on spectral range above 1350 nm with the strongest features (SWIR range). Hence, Exclusion of the spectral range from 350 to 1350 nm will not miss any major mineral components."*

**4- Have the authors explored the relationship between composition and size distribution? If so and if possible, a short discussion on it could be added.**

**Response:** Exploring the relationship between composition and size distribution was not in the scope of this research and that would take another paper to run the related experiments investigating the relationship between these two variables. Furthermore, this experiment will be tricky to perform for these samples given that they also contain urban materials that would affect the reflectance spectra of various size separates.